# Physicochemical Properties and Phosphorus Adsorption Capacity of Ceramsite Made from Alum Sludge

**Li Shi** [1,2]**, Xiaohong Zhao** [2,3,]*****, Yongxiang Cao** [1,2]**, Hongjuan Ma** [3,4] **and Xuejun Sun** [5]

1    Power China Northwest Engineering Corporation Limited, Xi'an 710065, China; shili@nwh.cn (L.S.); caoyongx@nwh.cn (Y.C.)
2    Shaanxi Union Research Center of University and Enterprise for River and Lake Ecosystems Protection and Restoration, Xi'an 710065, China
3    School of Civil Engineering, Chang'an University, Xi'an 710061, China; mahongjuan1129@outlook.com
4    Jinan Water Conservancy Construction Survey Design Research Institute Co., Ltd., Jinan 250100, China
5    Qujiang Water Works of Xi'an Water Supply Co., Ltd., Xi'an 710061, China; xj@163.com
*    Correspondence: xzhao@chd.edu.cn

**Abstract:** Alum sludge is an inevitable by-product from the water purification process, which had been applied as substrates in some constructed wetlands with good performance, especially for phosphorus (P) adsorption. The raw alum sludge is similar to a clay lump with an irregular shape, and there is a concern of it leaching into water. For better reuse, herein, some sludge was fired to produce alum sludge ceramsite (ASC) with a uniform spherical shape via a four-step process of kneading the sludge ball, air drying, preheating at 400 °C for 10 min, and firing at 600 °C for 5 min. Meanwhile, an air-dried alum sludge ball (adASB) was manufactured for comparison. The physicochemical properties and P adsorption ability of ceramsite were investigated subsequently. Through XRD and FT-IR tests, there was no obvious difference between ASC and adASB on the phase structure, but there was a certain amount of Al-OH group loss on the surface of ASC. The structure of ASC was still amorphous, similar to adASB, while ASC possessed more micropore structure and a bigger specific surface area than adASB. Adsorption experiments showed the P adsorption behaviors of ASC and adASB were much similar, and their adsorption kinetics were in accordance with the two-step adsorption kinetics rate equation and pseudo-second-order kinetics equation. The maximum adsorption capacities of ASC and adASB fitted by the Langmuir model were 1.66 mg/g and 1.89 mg/g, respectively. It should be pointed out that, compared with other adsorbents, the ASC produced in this study still had a greater ability to adsorb P. Therefore, ASC should have a great application potential for P removal in wastewater treatment in China.

**Keywords:** alum sludge; ceramsite; physicochemical property; phosphorus adsorption

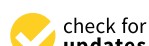


## 1. Introduction

With the process of urbanization and industrialization, more and more pollutants including organic and inorganic contaminants have been discharged into natural water bodies, resulting in high concentrations of BOD (biochemical oxygen demand), SS (suspended solids), and other types of pollutants in the receiving rivers, lakes, and reservoirs, particularly nitrogen and phosphorus [1]. Excessive nutrients can lead to eutrophication, the explosion in growth of algae, and eventually the deterioration of water quality, which has been a big environmental concern worldwide. It is generally believed that when the nitrogen content is greater than 0.2 mg/L and the phosphorus content is greater than 0.02 mg/L, it becomes eutrophic water. Thus, it is necessary and urgent to remove nutrients from natural water bodies, especially phosphorus which has the lowest threshold for the risk of eutrophication.

Over the past 20 years, different methods have been applied for eutrophication control, which can be classified to mechanical methods, chemical methods, and biological remedia-

tion methods. Although all of these approaches are useful, each of them has its drawbacks. For instance, secondary pollution was easily caused by chemical precipitation [2], while a long reaction period with stringent conditions is the distinct weakness of biological treatment [3]. In comparison, adsorption is a relatively convenient and reliable technique relying on an efficient adsorbent usually with a high cost [4]. As the main limiting element for eutrophication, how to remove P from water is the key issue to solve this problem. The maximum P adsorption capacity of diatomite modified by iron nitrate was 1.08 mg/g [5]. A novel diatomite adsorbent composited with a zeolitic imidazolate framework was developed, and its P sorption capacity reached 13.46, 13.55, and 13.95 mg/g at 25, 35, and 45 °C, respectively [6]. The activated carbon prepared from sugarcane bagasse can remove 22.64 to 99.27% P from 50 mL solution with increasing adsorbent dosages from 100 to 1500 mg, and the maximum P adsorption capacity was 1.13 mg/g [7]. The nutshell activated carbon modified by NaOH enhanced the adsorption of P to the maximum of 4.55 mg/g [8].

Alum sludge is the inescapable by-product from the water treatment works, and normally it is discharged by landfill. Therefore, the beneficial reuse of alum sludge is quite important for saving land and eliminating environmental risk. Much research has proved that alum sludge can be used as a low-cost P adsorption medium in wastewater treatment and had good performance in the lab experiments and some pilot tests [9]. Yang [10] reported that the maximum adsorption capacity of 3.5 mg-P/g was achieved when the pH of the synthetic P solution was 4.3, and the aging time had no significant effect on phosphorus adsorption of alum sludge [11]. Zhao [12] incorporated dewatered alum sludge cakes as main substrate into constructed wetlands receiving farm wastewater, indicating a high efficiency of 86.4 ± 6% on RP (reactive P) removal in a long-term trial. A novel alum sludge-based tidal flow constructed wetlands system was developed, and the average treatment efficiency for TP (total phosphorus) can reach 90% [13]. Compared with traditional adsorbents such as activated carbon, sulfonated coal, and diatomite used in wastewater treatment, alum sludge is drawing more and more attention due to its low cost and high efficiency for P removal [10].

However, as a kind of solid waste, there also existed some disadvantages for the reuse of alum sludge. As is well known, raw air-dried alum sludge is similar to clay rocks, with no definite shape and being quite loose, and there is a concern that the alum sludge could be eroded gradually by water and then eventually collapsed into run off.

Therefore, based on the idea of transforming alum sludge from waste into a resource and better reusing it in practice, the main subject of this study is to develop a treatment process for the production of alum sludge ceramsite (ASC) with a fixed shape and a certain mechanical strength, which simultaneously possesses a high capacity for P adsorption. It could be believed that this kind of ASC has the potential to be widely used as a novel adsorbent in wastewater treatment.

## 2. Materials and Methods

### 2.1. Preparation of ASC

The alum sludge was taken from a waterworks in Xi'an, China. Normally, the raw slum sludge is dark brown and pH-neural with high water content of around 80%. As tested before, TOC (total organic carbon) content in the air-dried alum sludge is roughly 15–20%, and others are inorganic substances. From the element test by XRF (1800, SHIMADZU Corporation, Kyoto, Japan), it was clear that the main components shown as oxides were $Al_2O_3$ and $SiO_2$, which are important for pottery. Certainly, the form of aluminum in alum sludge was not crystal $Al_2O_3$ but amorphous hydroxyl polymeric aluminum [10], but proper high temperature modification can beneficially transfer $Al^{3+}$ to $Al_2O_3$, which has an effect to support the skeleton of ceramsite. Therefore, alum sludge has the potential to produce ceramsite.

Based on the conventional process of ceramsite and characteristics of alum sludge, a three-step procedure of producing ASC was developed in this study. Firstly, due to the high moisture content of around 80%, the raw alum sludge can be kneaded by hand into the shape of ball without adding any extra water or other additions. Then, the alum sludge ball should be naturally air-dried in a well-ventilated place at room temperature (25 °C). Finally, when the weight of ball was stable, the air-dried alum sludge ball (adASB) was fired gradually in the muffle furnace (SX$_2$-2014, Boxun, Shanghai, China). The sintering process included (1) preheating for 10 min under about 400 °C, (2) heating with the temperature rising to 600 °C and kept sintering for 5 min, and (3) cooling in which the muffle was turned off and the ASC was cooled without being taken out for about 12 h.

### 2.2. P-Adsorption Tests

The P-containing solution used in the experiments was made by distilled water and a certain amount of $KH_2PO_4$ (AR). Two series experiments were implemented in laboratory, and the steps were described below:

(1) Adsorption kinetics: Some 2 g ASC (or 2 g adASB) was added to 100 mL of P-containing solution (the initial P concentration was 10 mg/L) in a series of Erlenmeyer flasks, and then pH was adjusted to 7 by using NaOH. Prepared flasks were placed on an orbital shaker to be shaken at 150 rpm. The samples were taken after shaking for 30, 60, 120, 240, 480, 720, 1440, 2160, 2880, and 3600 min, respectively, and filtered by a 0.45 μm millipore filter for P measurement.

(2) Isothermal adsorption experiment: Some 2 g ASC (or 2 g adASB) was added to 100 mL of P-containing solution in a series of Erlenmeyer flasks, and the initial P concentrations of the solution were 5, 10, 15, 20, 30, and 50 mg/L, respectively. Then, the pH of the solution was adjusted to 7. After shaking for 48 h, water samples were taken from every container to measure the concentration of P.

All the experiments were conducted at room temperature and replicated three times, and the results were shown in average values.

### 2.3. Analytical Methods

The P concentration was monitored by an ultraviolet–visible spectrophotometer (DR6000, Hach, Loveland, CO, USA) according to the standard method. To further study the characteristics of ASC and the impact of firing on alum sludge, ASC, and adASB were characterized by XRD (Bruker D8 Advance, Mannheim, Germany), FTIR (Nicolet iS50, Thermo, Waltham, MA, USA), SEM (Hitachi S-4800, Hitachi, Japan), and BET (Micromeritics ASAP2020, Norcross, GA, USA) methods. The XRD pattern can help to learn the phase structure of materials and the FT-IR spectrum to identify the changes of typical functional groups. Apparent morphology of ASC and adASB can be observed directly through the SEM. The specific surface area and porosity of materials can be examined with BET analysis.

### 2.4. P-Adsorption Models

There were three adsorption kinetics models to fit the experimental results, namely the pseudo-first order kinetic model, the pseudo-second order kinetic model, and the intra-particle diffusion model. The isotherm adsorption models adopted were Langmuir and Freundlich equations. All models were shown in detail in Table 1. In addition, the correlation coefficient ($R_2$) and four error functions were calculated to evaluate the fitting accuracy.

**Table 1.** The models adopted in this study.

| Name | Equation | Parameters |
| --- | --- | --- |
| Pseudo-first order kinetic model | $\log(q_e - Q_t) = \log q_e - K_1 t$ | $q_e$, $Q_t$: the amounts of adsorbed P at equilibrium and at time $t$ (min), respectively (mg/g); $K_1$: the first-order rate constant (min/L). |
| Pseudo-second order kinetic model | $\frac{t}{Q_t} = \frac{1}{K_2 q_e^2} + \frac{t}{q_e}$ | $K_2$: the second-order rate constant (g/mg min). |
| Intra-particle diffusion model | $Q_t = a + K_3 t^{1/2}$ | $K_3$: the intra-particle diffusion rate constant (min$^{-1}$). |
| Langmuir Isotherm | $\frac{C_e}{Q_e} = \frac{1}{K \cdot Q_m} + \frac{C_e}{Q_m}$ | $Q_e$: the mass of P adsorbed on adsorbent at equilibrium (mg/g); $C_e$: the equilibrium concentration of P solution (mg/L); $Q_m$: the maximum adsorption capacity (mg/g); $K$: Langmuir constant (l/mg), a measure of the affinity of the adsorbate for the adsorbent. |
| Freundlich Isotherm | $\log Q_e = \log K_f + \frac{1}{n} \log C_e$ | $K_f$: the Freundlich constant (l/g) related to the bonding energy; $1/n$: the heterogeneity factor. |

## 3. Results

### 3.1. ASC Product and Its Characterization

#### 3.1.1. Apparent Change

The procedure and apparent change of alum sludge in each stage of ASC producing process were shown in Figure 1. The raw alum sludge taken after sludge conditioning and dewatering process contained 85% water, which was brown in color, soft, and quite loose like clay mud. After being kneaded by hand, an alum sludge ball (ASB) was produced with a 10 mm diameter and a single weight of 1.1 ± 0.1 g. Through 3–4 days air drying, the alum sludge ball shrank significantly to dark brown adASB, losing about 80% of its weight, and the diameter was reduced to 7–8 mm with a single weight of 0.24 ± 0.05 g. In the firing process, approximately 45% of the weight of adASB was lost further, and the diameter of ASC shrank to 4–5 mm; the single weight reduced to 0.13 ± 0.01 g. The color of ASC changed to light brown, the same as ceramsite.

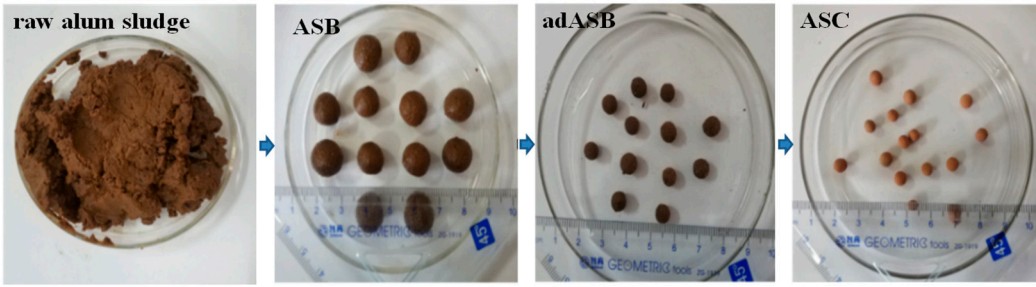

**Figure 1.** Each production stage of ASC and its apparent change.

#### 3.1.2. Microstructure of adASB and ASC

To further investigate the effect of firing process on microstructure of alum sludge, the analytic approaches including FT-IR, XRD, and BET were adopted, and the results are illustrated in Figure 2. Chemical groups are important factors affecting the chemical properties of materials. The chemical functional groups of ASC and adASB were determined by FT-IR analysis, and the results are shown in Figure 2A. It can be seen that the infrared spectra of asASB and ASC have obvious absorption peaks at wave numbers of 3438, 1635, 1418, 1087, 518, and 474 cm$^{-1}$. Through comparison of relevant studies and the standard spectrum, the greatest absorbance value appears at 3438 cm$^{-1}$, which was identified as the stretching

vibration of the OH group connected to Al, and the absorption peak at 1635 cm$^{-1}$ may be linked to the bonded OH or vibration of other organic functional groups. The absorbance value appears at the wave number of 1418 cm$^{-1}$ and may be the bending of Al-OH. As to the adsorption band appearing at 1087 cm$^{-1}$, it can be recognized as the Al-O group. The peaks of 518 and 474 cm$^{-1}$ may represent the stretch of O-Si-O [14,15].

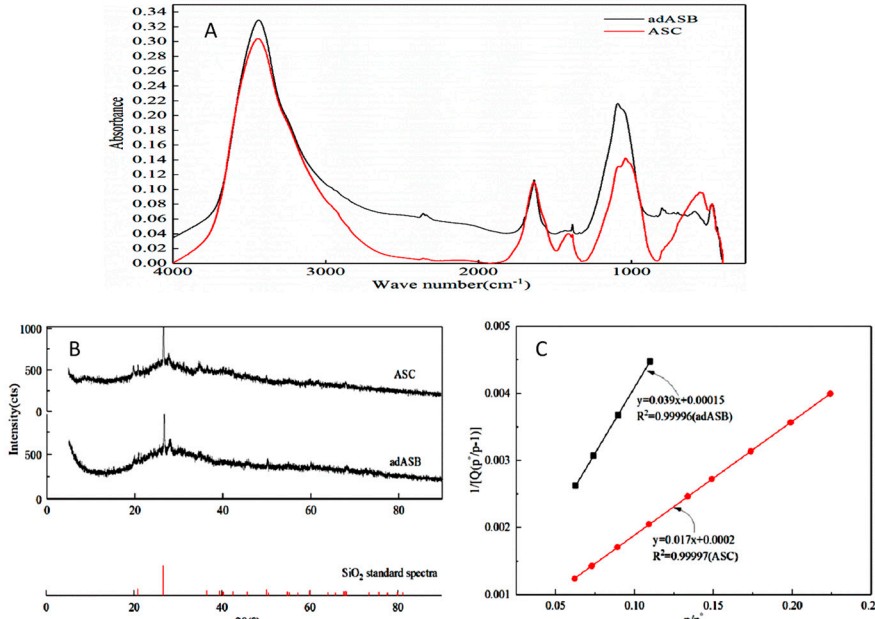

**Figure 2.** Microstructure of adASB and ASC: (**A**) FT-IR spectra, (**B**): XRD spectra, and (**C**): BET test.

Based on the FT-IR analysis above, the main chemical functional groups in both ASC and adASB can be inferred to be the $SiO_2$ and Al-OH groups which are related to the sediment carried by raw water and the residual coagulant of PAC. Compared with the spectra of ASC and adASB, the absorption peaks of sludge ceramsite at wave numbers of 3438, 1635, and 1418 cm$^{-1}$ were slightly weakened, probably due to the removal of coordination water molecules or hydroxyl groups during the firing process. On the contrary, at the wave number of 518 cm$^{-1}$, the absorption peak strength of ASC increased compared with that of adASB. This change may come from the loss of water molecules, then resulting in the increasing strength of the O-Si-O bond in ASC. On the whole, the firing process developed in this study just brought some loss of water and OH groups and no essential changes on either phase structure or chemical components.

The X-ray diffraction patterns of ASC and adASB were illustrated in Figure 2B. It can be seen clearly that their XRD characteristics are quite similar. The strongest and sharp peak corresponds to an angle of 26.6 degrees, which was identified as crystal $SiO_2$ by comparing the diffraction peaks in the standard diffraction spectra of typical $SiO_2$. The source of $SiO_2$ in alum sludge was mainly from the sediment brought by the raw water. There were no characteristics peaks of aluminum crystals in the spectra of the two samples, which indicated that the phase of aluminum with high content in ASC and adASB was still amorphous and the crystallization of aluminum did not occur during the firing process. The possible reason may be due to the low firing temperature and short firing time which were not enough to change the phase structure of alum sludge essentially. In short, except crystal $SiO_2$, the other elements in ASC or adASB are still amorphous, especially for the aluminum.

The BET plots of adASB and ASC are presented in Figure 2C. Through calculating from BET model, the results including $Q_m$, C, and BET surface area are shown in Table 2. The BET surface areas of adASB and ASC are 110.22 and 253.29 m$^2$/g, respectively, proving that the sintering process was helpful for the increase of BET surface area. During the

firing process, organic matter in the alum sludge was burned off to produce gases, and the escape of gases can lead to the formation of pores, thus resulting in the weight losing and increasing the BET surface area of ASC.

**Table 2.** BET analysis by using nitrogen adsorption isotherm.

| Materials | p/p° Range | Total Pore Volumes (cm$^3$/g) | Average Pore Diameters (nm) | BET Surface Area (m$^2$/g) | R$^2$ |
|---|---|---|---|---|---|
| adASB | 0.0626–0.1098 | 0.066 | 2.412 | 110.23 | 0.9996 |
| ASC | 0.06209–0.224 | 0.224 | 3.540 | 253.29 | 0.9997 |

The pore properties including pore type, pore size, and pore volume can also affect the adsorption capacity since the pores on the surface of the alum sludge ball provide the pathway for adsorbates, such as P. More importantly, any significant change in pore properties may result in the change of microorganism quantity adhering to alum sludge ball. Based on the BET analysis, the pore volume and pore size were determined. By using a single point adsorption method, the total pore volumes of ASC and adASB were 0.224 and 0.066 cm$^3$/g while the average pore diameters of ASC and adASB were 3.54 and 2.41 nm, respectively. With no doubt, the total pore volume and average pore size of alum sludge ball had increased greatly after sintering which could be a potential benefit for the adsorption ability of ASC.

The SEM images of adASB and ASC are shown in Figure 3. Obviously, there are several pores and fissures on their surfaces. When the magnification was 50 times, a certain number of cracks could be observed on the surface of ASC and adASB, which may be caused by the shrinkage of the sphere volume due to the evaporation of water during the drying process. It should be noted that the surface of ASC was smoother and there were fewer cracks compared with adASB, revealing that ASC was more solid and durable. When the magnification is 1000 times, the images showed that the surface of adASB and ASC are similar, very rough with irregular bulges, and some pores had a size distribution about 1~5 μm.

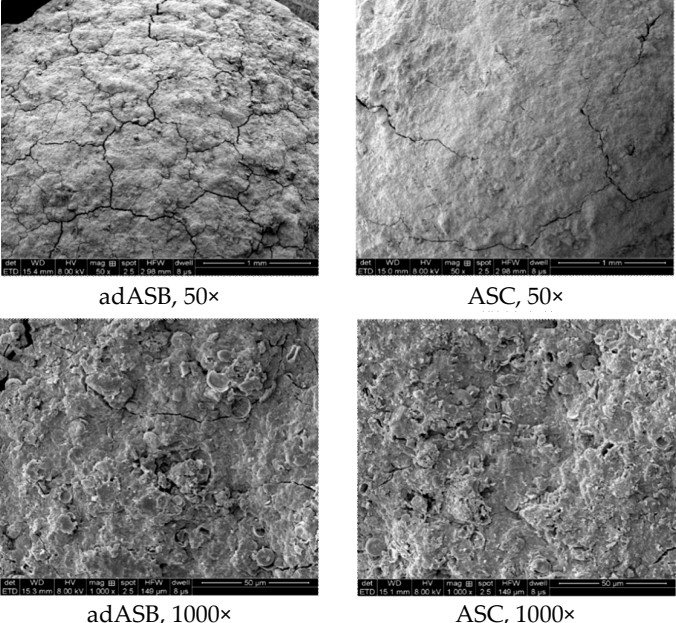

adASB, 50×                    ASC, 50×

adASB, 1000×                  ASC, 1000×

**Figure 3.** SEM images of adASB and ASC under different resolution ratios.

Overall, the test approaches used to analyze the physical–chemical characteristics revealed that the slight losing of OH group could bring an adverse effect on adsorption whereas increasing of the surface area and the pore volume (size) could bring some advantages. Therefore, the final change of adsorption ability should depend on their integrated effect.

### 3.2. Adsorption Behaviors on P Removal
#### 3.2.1. Adsorption Kinetics

The results of P adsorption kinetics of adASB and ASC are shown in Figure 4, which represented the relationship of time and residual P concentrations (A), the fitting results of the linear fitting of the pseudo-first order kinetic model (B), the pseudo-second order kinetic model (C), and the intra-particle diffusion model (D), respectively.

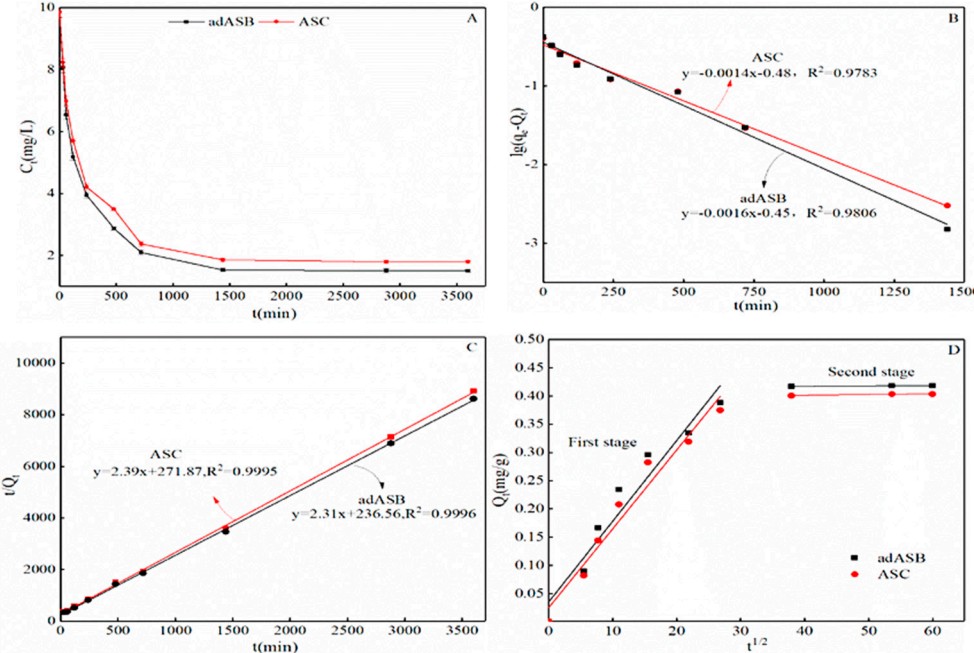

**Figure 4.** P adsorption behaviors of adASB and ASC: (**A**): the relationship of time and residual P concentration, (**B**): the fitting of pseudo-first order kinetic model, (**C**): the fitting of pseudo-second order kinetic model, and (**D**): the fitting of intra-particle diffusion model.

It is obvious that there was no big difference on the P adsorption behaviors between ASC and adASB. From Figure 4A, adsorption equilibriums were obtained after 48 h in two systems. The adsorption rate of P was sharp in the beginning and then dropped down gradually after 4 h, showing a trend of "adsorb quickly, equilibrate slowly". P removals in the adASB and ASC systems were 47.5 and 42.2% in 2 h and can reach to 78.3 and 76% in 12 h, respectively. After 48 h of the reaction, there was no more decrease in P content, indicating that P adsorption achieved equilibrium and 14.8 and 18.1% P were left in the adASB and ASC system, respectively. On the whole, the capacity of ASC on P adsorption was around 3.3% lower than that of adASB.

Generally, the chemical adsorption rate between P and metal elements such as Al, Fe, or Ca, which are active substances of adsorbents, is faster, while the physical adsorption rate between adsorbent and P is slower [16]. Therefore, the process of P adsorption by adASB and ASC all may include two stages: chemical adsorption and physical adsorption.

Theoretically, the control step of the adsorption process will be an intragranular diffusion if the $t^{1/2}$ and $Q_t$ relationship is a line across the zero point. From Figure 4D, $t^{1/2}$-$Q_t$ correlations of adASB and ASC are not linear in total, but the matched curves were divided into two different linear segments. As assumption above, the first stage nearly

across the zero point is chemical adsorption which was possibly controlled by intragranular diffusion, whereas the second stage may represent physical adsorption progress. In short, the adsorption behaviors of P by adASB and ASC were very similar and might be controlled by multiple steps [17].

The linear fittings of pseudo-first order kinetic model and pseudo-second order kinetic model are illustrated in Figure 2B,C, respectively. Additionally, Table 3 summarizes the computed values of $q_e$, $K_1$, $K_2$, $K_3$, and the correlation coefficient ($R^2$) in three adsorption kinetics models. The actual equilibrium adsorption capacities ($q_{esj}$) of adASB and ASC were 0.418 and 0.403 mg/L, while the computed $q_e$ values obtained by pseudo-first-order model of adASB and ASC were 0.357 and 0.332 mg/L, respectively. In contrast, the computed $q_e$ values achieved with pseudo-second-order model of adASB and ASC were a little bit higher, with 0.432 and 0.418 mg/L, respectively. These results revealed that the capacity of P adsorption by adASB was slightly higher than that of ASC, indicating that the ability of alum sludge for P removal had been reduced a little bit by sintering process. Based on correlation coefficients, the pseudo-second-order kinetic model for adASB and ASC ($R^2$ = 0.999–0.999) were more in line with the experimental data than the pseudo-first-order kinetic model ($R^2$ = 0.978–0.980), which indicated that there were rate-limiting steps on the adsorption of P for both two adsorbents, and the adsorption characteristics of P could be concluded as a kind of chemisorption for ASC and adASB.

**Table 3.** Summary of adsorption kinetic parameters on adsorption of P by different materials.

| Material | $q_{esj}$ | Pseudo-First-Order Kinetics | | | Pseudo-Second-Order Kinetics | | | Intra-Particle Diffusion | | |
|---|---|---|---|---|---|---|---|---|---|---|
| | | $q_e$ | $K_1$ | $R^2$ | $q_e$ | $K_2$ | $R^2$ | $R^2$ (First Stage) | $K_3$ | $R^2$ (Second Stage) |
| adASB | 0.418 | 0.357 | 0.0016 | 0.980 | 0.432 | 0.022 | 0.999 | 0.958 | 0.014 | 0.924 |
| ASC | 0.403 | 0.332 | 0.0014 | 0.978 | 0.418 | 0.020 | 0.999 | 0.952 | 0.014 | 0.921 |

### 3.2.2. Adsorption Isotherm

The results of adsorption isotherms of ASC and adASB were illustrated in Figure 5. As shown in Figure 5A, the residual P concentrations ($C_t$) after 48 h reaction (equilibrium concentration) increased along with the increasing initial P contents ($C_0$, from 5 to 50 mg/L with different intervals), and the higher initial P contents, the greater difference of residual P between the ASC and adASB system. When the initial P concentration was 5 mg/L, the residual P concentrations in the adASB and ASC systems were 0.72 mg/L and 0.79 mg/L, respectively, whereas when the initial P was 50 mg/L, the residual P of the adASB and ASC adsorption system were 22.56 mg/L and 24.84 mg/L, respectively. Accordingly, the amounts of P removed by ASC were 0.21 and 1.26 mg/g under the initial P of 5 and 50 mg/L, respectively, indicating the initial P concentrations had a certain effect on the adsorption capacities of both ASC and adASB.

Langmuir and Freundlich adsorption isotherm models were adopted for the fitting of experiments data, and the results are shown in Figure 5B,C. The parameters such as $q_e$, $K$, etc., are summarized in Table 4. Based on the values of correlation coefficient $R^2$, the Langmuir model was more in line with the experimental data (both adASB and ASC) than the Freundlich model. The theoretic maximum adsorption capacities represented by $Q_m$ were 1.66 mg/g for ASC and 1.89 mg/g for adASB obtained from Langmuir fitting, which means that the firing process may have a slight negative effect on the adsorption characteristics of alum sludge.

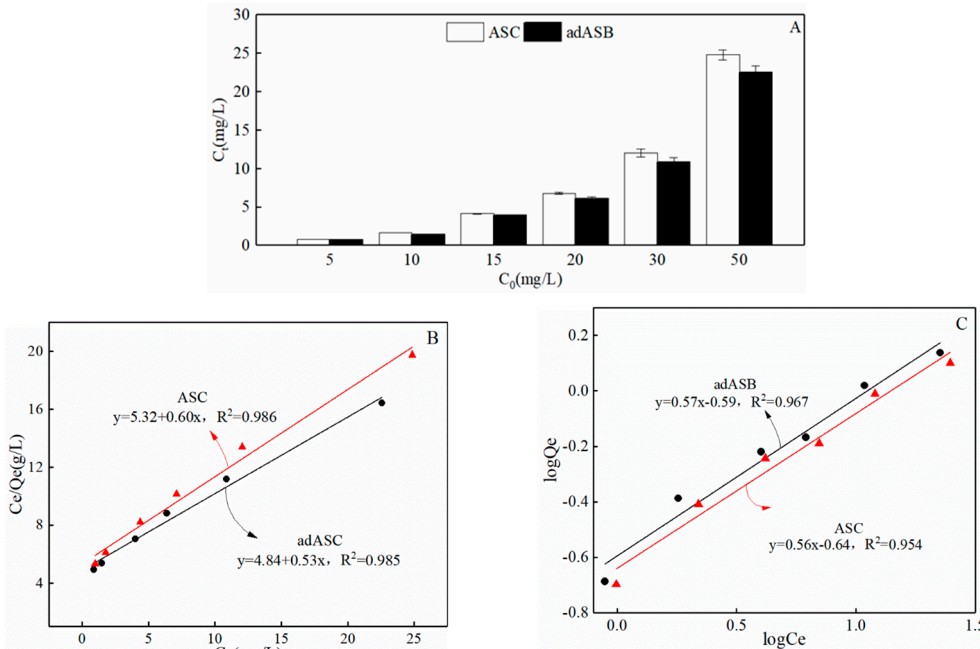

**Figure 5.** P-adsorption isotherms of adASB and ASC: (**A**): the relationship of initial P concentration and residual P after 48 h reaction; (**B**): the fitting of Langmuir model; and (**C**): the fitting of Freundlich model.

**Table 4.** Summary of adsorption isotherm parameters on adsorption of P by different materials.

| Material | Langmuir | | | Freundlich | | |
|---|---|---|---|---|---|---|
| | $Q_m$ | $K$ | $R^2$ | $K_f$ | $1/n$ | $R^2$ |
| adASB | 1.89 | 0.11 | 0.985 | 0.257 | 0.57 | 0.967 |
| ASC | 1.66 | 0.11 | 0.986 | 0.229 | 0.56 | 0.954 |

## 4. Discussion

### 4.1. The Firing Procedure and Features of ASC

According to the firing procedure and inner reactions that occurred, there are two kinds of ceramsites: sintered ceramsite and expended ceramsite [18]. Normally, the sintered ceramsite is more compact with polycrystalline oxides and has high mechanical strength under higher firing temperature and mixture of suitable supplements, which could be used as building material, whereas the specific surface area and the porosity of expended ceramsite must be improved significantly via aerogenesis and bloating during the firing process. Therefore, except for the raw materials having a certain recipe, the firing temperature must be controlled precisely, normally higher than 1100 °C. If the recipe of raw materials is 40–68% $SiO_2$, 12–18% $Al_2O_3$, 5–10% $Fe_2O_3$, and 2.5–7% $K_2O$ and $Na_2O$, there is no need of expanding aid [19]. Alum sludge was used as main material added with bentonite to make ceramsite under 1050 °C firing for 10 min [20]. Similarly, mixed with clay, alum sludge can be changed to ceramsite by firing at 1150 °C for 8–10 min [21,22]. Since the calcination temperature has a great influence on the surface area and the active functional groups of the ceramsite which affect the adsorption performance of P, there were several studies trying to produce alum sludge pellets without any additive under lower temperatures ranging from 200 to 800 °C. Zheng [23] fired alum sludge balls at 200, 400, 600, and 800 °C for 3 h, respectively, and found that the pellets produced at 400 and 600 °C have higher P adsorption capacity. In addition, higher contents of Si and Al in material required higher firing temperature [24]. Alum sludge adopted in this trial has high Si and Al content,

needing high firing temperature ordinarily which is high in energy consumption and may result in the collapse of internal holes and damage of activated groups [24].

The phase structure of the alum sludge played an important role in its adsorption ability. The reaction of amorphous aluminum in alum sludge with phosphate was the main mechanism for P adsorption by alum sludge, and compared with the crystal aluminum, amorphous aluminum can provide a larger specific surface area and a stronger ability of P adsorption [10]. Meanwhile, the functional group containing alum sludge was also closely related to the P adsorption capacity of alum sludge. The OH group plays an important role in the ligand exchange process, because phosphate ions can be adsorbed by ligand exchange with -OH on the surface of alum sludge [11]. Therefore, aiming to neither destroy functional groups for P adsorption of alum sludge nor require high amounts of energy and resources, the procedure of low-temperature firing was developed in this paper.

It should be mentioned that preheating is also a key process in addition to the firing, which can eliminate the explosion induced by dramatic change of temperature and prepare for aerogenesis. Generally, the preheating temperatures ranged from 300 to 500 °C. In this trail, preheating at 400 °C for 10 min was implemented and firing temperatures of 400, 500, 600, 700, and 800 °C were tried, respectively. The results showed that, at 400 or 500 °C, the obtained pellets were an uneven dark color and there was rupture of pellets at 700 and 800 °C. Nevertheless, hard ASC with a perfect spherical shape and light brown color can be produced under 600 °C firing for only 5 min, which was selected consequently.

Based on the micro-structure analysis via the FT-IR, XRD, BET, and SEM methods and the P adsorption experiments, there was little difference of phase structure or chemical groups between ASC and adASB. The maximum adsorption capacity of ASC fitted by the Langmuir model is 1.66 mg/g, close to the 1.85 mg/g of adASB, indicating that the firing process we used had a little effect on the physical and chemical characteristics of alum sludge.

In practice, clay ceramsite and shale ceramsite are being used widely worldwide [25]. However, clay and shale are non-renewable resources, and the overexploitation of them has a potential risk to the environment. Thus, the reuse of municipal sludge including alum sludge from waterworks and activated sludge from wastewater treatment plants as a raw material for making ceramsite is more and more attractive for researchers. Compared with activated sludge, alum sludge is more like clay, and there are few harmful pollutants such as heavy metals and pathogenic bacteria suitable to make ceramsite at a low temperature. Moreover, the previous study proved that there was little leaching of aluminum when alum sludge was used in wastewater treatment [26]. Hence, producing ASC can not only realize the reuse of alum sludge but also can be used as a novel low-cost adsorbent for P removal in practice.

### 4.2. Prospects of Using ASC in China

In China, sludge from drinking water treatment is estimated to be approximately $200 \times 10^4$ t in dry solids per year [27]. Due to the economy's development and rapid urbanization, this amount is increasing annually. If a landfill site is 25 km away from waterworks, the cost including transporting and landfill will be 70–100 RMB per ton [28], which should be a big burden for the waterworks. Meanwhile, landfills waste and damage the valuable and limited resource of land. Therefore, how to rationally reuse alum sludge is still a big challenge in China. Although in an infant stage, this study provides a promising method of transforming raw alum sludge to ASC, which is low-cost and easy control. Compared with other firing procedures, there is no additive such as extra water, binders, or fluxing agent, and the firing temperature and time are lower, significantly saving energy and investment. In respect of P adsorption ability, ASC showed a similar capacity as asASB. As shown in Figure 6, the final product ASC with a fixed shape and a certain mechanical strength could be used either as substrates for growing plants or a as P adsorbent used in wastewater treatment. Compared with the employment of alum sludge directly, it is convenient to recycle ASC and possibly recover P from it. In total, reusing waste alum

sludge to treat wastewater and adsorb and recover P is not only a win–win method but is also in line with sustainable development of China.

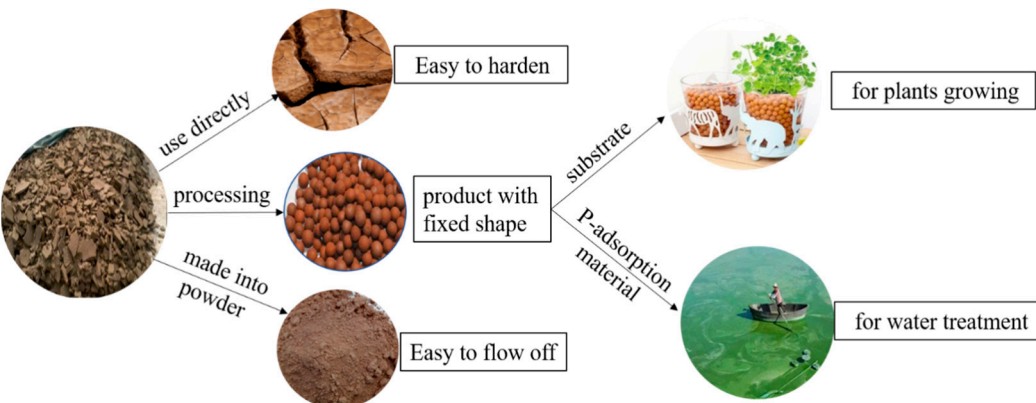

**Figure 6.** The potential use of ASC.

It should be pointed out that this trial only developed the procedure of making ceramsite from alum sludge at lab-scale. Scalability and practicality of producing ASC on a larger scale are a complex issue with a long footprint. Hence, further exploration should be carried out at a big scale to ensure that the procedure and products are reliable and workable in practice. Thereafter, some applications such as using ASC as a substrate in a floating bed for eutrophication control or as a medium in the constructed wetland should be tried in the future.

## 5. Conclusions

ASC made from alum sludge in this study showed a similar P-adsorption ability compared with that of adASB, and its adsorption kinetics are in accordance with the two-step adsorption kinetics and pseudo-second-order kinetics equation. The maximum adsorption capacity of ASC fitted by the Langmuir model is 1.66 mg/g, close to the 1.85 mg/g of adASB, which means the firing process has a slight effect on P adsorption capacity. The XRD and FT-IR results revealed that the phase structure and functional groups on the surface of the alum sludge changed slightly over the firing process, expressed as the loss of -OH and Al-OH, which may be the reason of reduce of the adsorption capacity of ASC. The morphological appearance observed by SEM shows that the surface of ASC is smoother than that of adASB, but the specific surface areas of ASC and adASB were 253.29 and 110.23 m$^2$/g, respectively, from BET test. In addition, the pore size and pore volume of ASC increased simultaneously. In brief, after the firing process, alum sludge can be changed to ASC with fixed shape and good ability of P adsorption, which could have a potential to be used as low-cost absorbent in wastewater treatment.

**Author Contributions:** Conceptualization, X.Z. and L.S.; Funding acquisition, L.S. and X.S.; Investigation, X.Z. and Y.C.; Methodology and formal analysis, X.Z.; Supervision, L.S. and Y.C.; Writing—original draft, H.M.; Writing—review and editing, X.Z. All authors have read and agreed to the published version of the manuscript.

**Funding:** This research was funded by the Key Laboratory of Subsurface Hydrology and Ecological Effects in Arid Region, Ministry of Education (Chang'an University) (No. 300102291501) as well as Shaanxi Union Research Center of University and Enterprise for River and Lake Ecosystems Protection and Restoration with funding from Power China Northwest Engineering Corporation Limited, Xi'an 710065, China (No. 220228220662).

**Data Availability Statement:** Not applicable.

**Conflicts of Interest:** The authors declare no conflict of interest.

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
