# Peer review of "Physicochemical Properties and Phosphorus Adsorption Capacity of Ceramsite Made from Alum Sludge"

_water, doi:10.3390/w15132427_

Round 1

Reviewer 1 Report

The paper discusses the issue of eutrophication caused by the discharge of pollutants into natural water bodies and emphasizes the need for removing excess nutrients, particularly nitrogen and phosphorus. The authors propose the use of alum sludge as a low-cost and efficient adsorbent for phosphorus removal in wastewater treatment. They further present a study on the development of alum sludge ceramsite (ASC) with fixed shape, mechanical strength, and high phosphorus adsorption capacity

While the article provides some valuable insights into the potential use of alum sludge ceramsite for phosphorus removal some points need to be addressed:

- the article lacks a proper introduction to the topic of eutrophication and the significance of phosphorus removal in wastewater treatment

- the article briefly mentions different methods for eutrophication control but fails to provide a comprehensive review of the existing research in the field

- I find the presentation of results a little confusing. The authors discuss the physical and chemical characteristics of ASC without clearly linking them to the adsorption capacity for phosphorus. The relationship between the structural changes of ASC and its phosphorus removal performance should be explicitly explained.

- It could be worth to do a short comparison with alternative adsorbents. You mention that alum sludge is drawing more attention due to its low-cost and high efficiency for phosphorus removal. Could you provide a thorough comparison with other commonly used adsorbents, such as activated carbon or diatomite. A comparative analysis would provide a better understanding of the advantages and limitations of ASC.

- The study focuses on the development and characterization of alum sludge ceramsite in lab experiments. Some comments on scalability and practicality of producing ASC on a larger scale should be addressed.

In my opinion addressing these issues would significantly improve the article's scientific overtone and readability.

Some minor editing corrections are necessary e.g.:

- references to literature in the whole text are connected with words, the space is required
- in the list, in chapter 2.1, start all elements with small letters   
- there are some cases where words are connected e.g.: t ^1/2-Qtcorrelations, K3and

Reviewer 2 Report

This manuscript investigated the physico-chemical properites of P adsorption capacity of ceramsite made from alum sludge and compare it with conventional air-dry alum sludge ball. Overall, the research is interesting, but there are main concerns about the abstract and introduction section. Since the whole paper is compare ASC with adASB, there is lack of information about adASB. More detailed comments can be found below.

(1) Title, capacity not Ca-Pacity, please revise it.

(2) In the abstract, it is a bit sudden to compare the ASC with adASB? It should be one sentence to talk about the adASB. Is it a conventional alum sludge treatment method?

(3) There is lack of information about the adASB in the introduction as well. In the aim section, the author also did not mention adASB at all.

(4) 2.1 The TOC content in the air-dried alum sludge is roughly 15-20%. This is quite strange, why the unit to TOC is % instead of mg/L.

(5) In the conclusion section, the author mentioned “The maximum adsorption capacity of ASC fitted by Langmuir model is 1.66 mg/g, close to the 1.85 mg/g of adASB”. Since the fire process has adverse effect on the P adsorption capacity, why the author developed this method?

(6) The specific surface areas of ASC and adASB were 253.29 and 110.23 m2/g respectively from BET test. How did the author explain that the ASC has double specific surface area but has lower P adsorption capacity. 

The English can be improved after the manuscript has been thoroughly revised. 

Round 2

Reviewer 2 Report

The author has well addressed all my concerns.